# Early Multiphasic HBV Infection Initiation Kinetics Is Not Clone-Specific and Is Not Affected by Hepatitis D Virus (HDV) Infection

**DOI:** 10.3390/v11030263

**Published:** 2019-03-15

**Authors:** Masataka Tsuge, Takuro Uchida, Kevin Walsh, Yuji Ishida, Chise Tateno, Upendra Kumar, Jeffrey S. Glenn, Christopher Koh, Theo Heller, Susan L. Uprichard, Harel Dahari, Kazuaki Chayama

**Affiliations:** 1The Program for Experimental & Theoretical Modeling, Division of Hepatology, Department of Medicine, Loyola University Medical Center, Maywood, IL 60153, USA; tsuge@hiroshima-u.ac.jp (M.T.); kwalsh14@luc.edu (K.W.); upendra2154@gmail.com (U.K.); suprichard@luc.edu (S.L.U); hdahari@luc.edu (H.D.); 2Department of Gastroenterology and Metabolism, Graduate School of Biomedical & Health Sciences, Hiroshima University, Hiroshima 734-8551, Japan; tuchida@hiroshima-u.ac.jp; 3Research Center for Hepatology and Gastroenterology, Hiroshima University, Hiroshima 734-8551, Japan; yuji.ishida@phoenixbio.co.jp (Y.I.); chise.mukaidani@phoenixbio.co.jp (C.T.); 4Natural Science Center for Basic Research and Development, Hiroshima University, Hiroshima 734-8551, Japan; 5Liver Diseases Branch, NIDDK, NIH, Bethesda, MD 20892, USA; christopher.koh@nih.gov (C.K.); theoh@intra.niddk.nih.gov (T.H.); 6PhoenixBio Co., Ltd., Higashi-Hiroshima, Hiroshima 739-0046, Japan; 7Department of Medicine, Division of Gastroenterology and Hepatology, Stanford University School of Medicine, Stanford, CA 94305, USA; jeffrey.glenn@stanford.edu

**Keywords:** HBV, HDV, human hepatocyte chimeric mice, viral-host interactions, HBV/HDV co-infection

## Abstract

**Backgrounds and Aims:** We previously demonstrated that serum hepatitis B virus (HBV) DNA in HBV infected humanized mice exhibited a highly dynamic multiphasic kinetic pattern from infection initiation to steady-state. Here, we investigated whether this pattern is consistent across different HBV clones or in the presence of hepatitis D virus (HDV) co-infection. **Methods:** We analyzed early serum viral kinetics using 26 HBV genotype C (GtC) mono-infected mice [clones: PXB, Hiroshima GtC CL4 (CL4) and Hiroshima GtC CL5 (CL5)] and four HBV CL4/HDV genotype one co-infected mice. **Results:** The HBV kinetics observed with clones CL4 and CL5 were similar to that previously defined in HBV PXB infected mice. Additionally, no significant differences in HBV DNA levels were observed between HBV mono-infected and HBV/HDV co-infected mice through 4 weeks post-inoculation (p.i.). However, HBV DNA levels at 6 weeks p.i. in HBV/HDV co-infected mice were significantly lower than those in HBV mono-infected mice (*P* = 0.002), consistent with HDV suppression of chronic HBV. **Conclusions:** HBV infection initiation is multiphasic across multiple viral clones and is not altered by HDV co-infection. The latter suggests that higher HDV titers (>8 log IU/mL) and/or longer duration of HDV infection might be needed to trigger HDV-induced suppression on HBV.

## 1. Introduction

Recently, we [1] showed that serum hepatitis B virus (HBV) DNA in HBV infected humanized mice exhibited a highly dynamic multiphasic kinetic pattern from infection initiation to steady state. Five of the 7 phases observed occurred within the first 8 days post-inoculation (p.i.) followed by an extended serum HBV DNA amplification phase before steady state was reached between 35–42 days p.i. Here, we examined whether the observed multiphasic HBV kinetic pattern is consistent among different HBV genotype C (GtC) clones and whether co-infection with the hepatitis D virus (HDV), which has been reported to suppress chronic HBV in humans [2] and humanized mice [3], might affect the observed HBV initiation kinetics.

## 2. Materials and Methods

### 2.1. Preparation of the Viral Inocula

Inocula consisted of serum from virus infected humanized mice. All sera containing HBV GtC were positive for HBs and HBe antigens with high-level HBV DNA. The inoculum containing PXB clone (AB246345) was obtained from HBV infected mice which were infected with serum from an HBV GtC infected patient in Nagoya City University. The inocula containing Hiroshima GtC CL4 clone (CL4 clone; MH887433) or Hiroshima GtC CL5 clone (CL5 clone; MH891502) were derived from HBV GtC infected patients at Hiroshima University. None of these clones contain any notable mutations.

The serum containing HDV genotype one was obtained from a patient at the National Institutes of Health (NIH). The serum was positive for HBs antigen with high-level HDV RNA, but HBV DNA was undetectable. The serum containing HDV was inoculated into humanized mice that were already infected with HBV CL4 to generate HBV/HDV superinfected mice with high titers of both HBV and HDV. Serum containing the equivalent titers of HBV and HDV was chosen as the inoculum for generating HBV/HDV co-infected mice (described in Experiment 4 below). All inocula were stored in liquid nitrogen until use.

### 2.2. Human Hepatocyte Chimeric Mouse Experiments

Preparation of hemizygous cDNA-uPA^+/−^/SCID^+/+^ mice and transplantation of human hepatocytes were performed as described previously [4]. In all cases, the human hepatocyte repopulation was 90% or greater. Mice were inoculated as indicated below by tail vein injection. HBV serum DNA kinetics from 30 humanized mice from four independent experiments were included. Experiment 1: One representative chimeric mouse from Ishida et al. [1], transplanted with human hepatocytes (lot: 5YM), was inoculated with 10^6^ HBV genome equivalents (GE) (PXB clone). Experiment 2: Nineteen humanized mice from Tsuge et al. [5] and unpublished data, transplanted with human hepatocytes (lot: 2YF), were inoculated with 10^6^ HBV GE (CL5 clone). Experiment 3: Six humanized mice (unpublished data), transplanted with human hepatocytes (lot: 2YF), were inoculated with 10^6^ HBV GE (CL4 clone). Experiment 4: Four humanized mice, transplanted with human hepatocytes (lot: 2YF), were inoculated simultaneously with 10^6^ GE of both HBV (CL4 clone) and HDV. Serum samples were obtained from mice at several time points and stored at −80 °C until use.

The experimental protocols utilized met the ethical guidelines of the Declaration of Helsinki, were approved by the Hiroshima University Ethical Committee (Approval ID: D08-9) and were performed in accordance with the guidelines of the local committee for animal experiments by Hiroshima University Committee for Recombinant DNA Experiments and Animal Experiments (Approval ID: 27-113-6 and A17-106, 26 July 2017 and 3 October 2017, respectively). All animals received humane care with both infection and serum sampling performed under anesthesia.

### 2.3. Quantification of Serum HBV DNA 

Quantitative analysis of HBV DNA was performed by real-time PCR as previously described. Briefly, serum HBV DNA was quantified from 10 µL of mouse serum by real-time PCR using the TaqMan PCR System (Roche Diagnostics, Tokyo, Japan). The lower quantitation limit of this assay is 3.5 Log copies/mL.

### 2.4. Quantification of serum HDV RNA 

RNA was extracted from serum samples using SepaGene RVR (EIDIA Co., Ltd., Tokyo, Japan) and reverse-transcribed with a random hexamer and a reverse transcriptase (ReverTraAce; TOYOBO, Osaka, Japan) according to manufacturers’ instructions. After the RT reaction, HDV cDNA was quantified by real-time PCR using the 7300 Real-Time PCR System (Applied Biosystems, Foster City, CA, USA). Amplification was performed in a 25 μL reaction containing 12.5 μL SYBR Green PCR Master Mix (Applied Biosystems), 5 pmol of forward primer, 5 pmol of reverse primer, and 1 μL of cDNA solution. After incubation for 2 min at 50 °C, the sample was denatured for 10 min at 95 °C, followed by a PCR cycling program consisting of 40 cycles of 15 s at 95 °C, 30 s at 55 °C, and 60 s at 60 °C. The lower quantitation limit of this assay is 5.0 Log copies/mL.

### 2.5. Statistical Analyses

The differences between groups were examined for statistical significance using the Mann-Whitney U test and Kruskal Wallis test. Statistical analysis was performed using SPSS○R ver. 17.0 (SPSS Inc., Chicago, IL, USA). All *P* values less than 0.05 by two-tailed test were considered significant.

## 3. Results and Discussions

The HBV infection kinetics observed for HBV clones CL5 (Exp2) and CL4 (Exp3) were similar to that in PXB (Exp1), indicating reproducible kinetic patterns across all three HBV clones (Figure 1A). Interestingly, HDV co-infection also had no effect on the HBV kinetic pattern observed during the first 4 weeks (Figure 1A) with no significant difference in serum HBV levels at weeks 1, 2, 3, or 4 p.i. regardless of HBV clone or HDV co-infection (Figure 1B). However, HBV DNA levels at 6 weeks p.i. in HBV/HDV co-infected mice were significantly lower than those in HBV mono-infected mice (*P *= 0.002) (Figure 1B). 

While both we (Figure 1) and others [3] have observed that HBV DNA levels are suppressed later in infection in the presence of HDV, the exact time of that suppression and relationship to HDV infection level is unclear. Specifically, the study by Giersch et al. [3] found that at 8 weeks, when mean HDV-RNA level reached 4.5 × 10^7^ copies/mL, there was significantly higher innate signaling in their co-infected mice which corresponded with suppressed serum HBV DNA. In our study, despite the fact that HDV RNA levels had already reached 10^8^ copies/mL by 4 weeks p.i., no suppression of serum HBV DNA was observed. In the absence of intracellular interferon stimulated genes or cytokines analysis, the lack of serum HBV DNA suppression in our co-infected mice at 4 weeks suggested that perhaps enhanced innate signaling was not yet present, and thus that peak innate signaling might be associated with the kinetics of infection rather than the absolute levels of HDV RNA at any specific time. Although host innate immune responses to viral infections are generally thought to occur rapidly, this highlights that some innate responses may need to be activated via inductions of several intracellular signal pathways [5], perhaps related to not only higher viral titers, but longer duration of infection.

This study has several limitations. First, while 26 mono-HBV infected mice were analyzed, only four HBV/HDV co-infected mice were investigated; thus it will be necessary to study more co-infected mice to independently verify the lack of difference in early kinetics between HBV mono-infected and HBV/HDV co-infected mice. Second, the sampling time points among the four experiments were not always identical because these experiments were originally designed independently (e.g., the comparison of HBV DNA levels at week three between mono- and co-infected mice (Figure 1B) were based on the data from only two mono-infection experiments).

Importantly, however, the sampling was frequent enough to allow comparison of kinetic patterns, and thus we were able to clearly observe the same reproducible multiphasic HBV infection initiation kinetic patterns (i.e. days 1–28 p.i.) independent of HBV clone and HDV co-infection. While delayed enhancement of HDV-induced innate signaling provides a reasonable explanation for why we observed no difference in early HBV infection kinetics between HBV mono-infected and HBV/HDV co-infected mice, this speculation, as well as the mechanistic-basis of the slow HDV-induced innate responses, warrants further investigation.

## Figures and Tables

**Figure 1 viruses-11-00263-f001:**
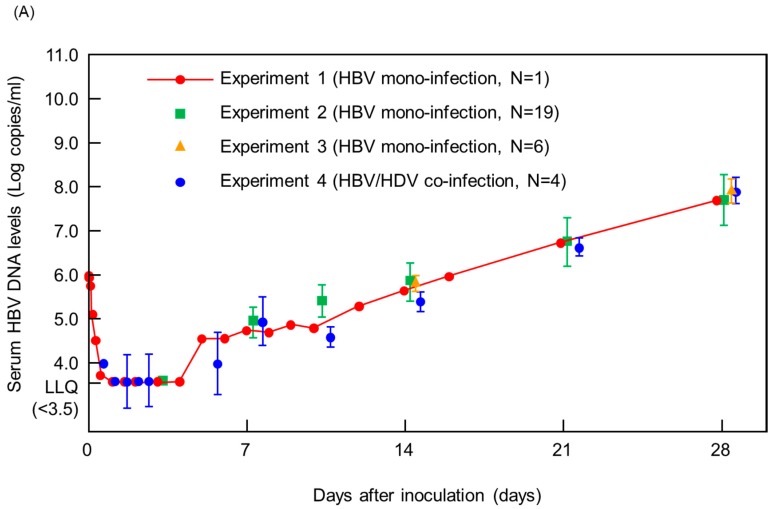
Comparison of HBV infection initiation kinetics in human hepatocyte chimeric mice. (**A**) Average hepatitis B virus (HBV) load with standard deviation from days 0–28 post inoculation (p.i.) in each experiment. (**B**) Comparison of serum HBV DNA levels between each experiment at indicated time points. Statistical analyses of HBV DNA levels were performed between HBV mono-infected and HBV/HDV co-infected mice (Experiment 1, 2, and 3 vs. Experiment 4) by Kruskal Wallis test. *P*-values <0.05 indicated that HBV DNA levels were statistically significant among the experiments. N.A.: not available because of the original experimental sampling design, which was not identical among the experiments.

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
