# Peer review of "Early Multiphasic HBV Infection Initiation Kinetics Is Not Clone-Specific and Is Not Affected by Hepatitis D Virus (HDV) Infection"

_viruses, 2019, doi:10.3390/v11030263_

Reviewer 1 Report

This short communication reports the results on HBV and HDV kinetics of replication in a mouse model during the early stages of infection. The study is interesting, however, there are some points to be clarified:

1.The authors should specify HBV and HDV genotypes of clones used in the experiments.

2. Previous studies have shown that HBsAg interacts with HDAg, a step crucial for HDV morphogenesis. Mutations in HBsAg can affect this phenomenon. In this light, Have the authors information on HBsAg sequence from the clones used in the experiments?

3. The authors should soften the conclusions on innate immune response. Although reasonable, the study evaluated only HBV and HDV kinetics and did not investigate immunological markers of innate immune response.

4. In the title, the authors should specify that HDV does not affect HBV kinetics of replication at the early stages of infection.

Author Response

Please find the attached file containing our responses to reviewers' comments.

Reviewer 2 Report

The paper by Tsuge M et al. is an interesting analysis of the early kinetics of HBV infection in humanized mice using different HBV clones and in presence or not of HDV coinfection. The study is innovative but limited by the exiguous number of experiments with HBV/HDV coinfection. This permits to draw only preliminary observations, deserving further confirmations.

In details, the paper reports together the data from 4 different experiments: 3 including HBV monoinfected mice and whose results have been already published, with the exception of the last one and 1 analysing HBV/HDV coinfected mice. The major limitation of the study is the low number of HDV-coinfected mice (4) included in this paper, that makes the conclusions not so strong. At this regard, the authors should stress the limited number of data obtained from coinfected mice in the discussion section, that could be responsible for the lack of difference in HBV-DNA kinetics in comparison to HBV monoinfected mice.

In addition, the authors analysed all together the data from 3 different experiments of monoinfection but it is not clear if all the experimental conditions were completely identical. It would be beneficial for a right comparison that any eventual difference in experimental conditions is clarified in the text.

Furthermore, the authors should address the following comments:

Could the authors give some more details on the different clones used in the 3 experiments of HBV monoinfection? Are there any genetic differences in the HBV backbone of the 3 clones?

Could the authors specify the timing used for HDV coinfection? How many hours/days after HBV infection did the authors inoculated HDV into humanized mice? This timing could be relevant for the influence of HDV on HBV kinetics. The authors should discuss this aspect more in details.

Regarding Figure 1. Comparison of HBV infection initiation kinetics in human hepatocyte chimeric mice.

1A The figure is not easily readable. Could the authors try to differentiate better the lines indicatinf the 4 different experiments?

1B Could the authors explain and report in the legend the reason why some data are not available (n.a. i.e. HBV-DNA in experiment 3 at 3 weeks and 1 at 6 weeks).

Secondly, the comparison of HBV-DNA levels at week 3 between mono- and coinfected mice are based only on the data of 2 experiments of mono-infection (including experiment 1 based only on 1 mouse). The authors should clarify this and discuss the limitation of this observation.

Author Response

Please find the attached file containing our responses to reviewers' comments.

Round  2

Reviewer 1 Report

The authors have properly answered to the points raised in my previous review